# p53 Signaling on Microenvironment and Its Contribution to Tissue Chemoresistance

**DOI:** 10.3390/membranes12020202

**Published:** 2022-02-09

**Authors:** Leonel Cardozo de Menezes e Souza, Anderson Faletti, Carla Pires Veríssimo, Mariana Paranhos Stelling, Helena Lobo Borges

**Affiliations:** 1Instituto de Ciências Biomédicas, Universidade Federal do Rio de Janeiro, Rio de Janeiro 21941-902, Brazil; leonel.cardozo20@gmail.com (L.C.d.M.e.S.); 7andersonfaletti7@gmail.com (A.F.); verissimocp@gmail.com (C.P.V.); 2Instituto Federal de Educação, Ciência e Tecnologia do Rio de Janeiro, Rio de Janeiro 20260-100, Brazil; mariana.stelling@ifrj.edu.br

**Keywords:** tumor microenvironment, p53 signaling, secretome, extracellular vesicles, drug resistance, mutant p53, cell-nonautonomous function

## Abstract

Chemoresistance persists as a significant, unresolved clinical challenge in many cancer types. The tumor microenvironment, in which cancer cells reside and interact with non-cancer cells and tissue structures, has a known role in promoting every aspect of tumor progression, including chemoresistance. However, the molecular determinants of microenvironment-driven chemoresistance are mainly unknown. In this review, we propose that the *TP53* tumor suppressor, found mutant in over half of human cancers, is a crucial regulator of cancer cell-microenvironment crosstalk and a prime candidate for the investigation of microenvironment-specific modulators of chemoresistance. Wild-type p53 controls the secretion of factors that inhibit the tumor microenvironment, whereas altered secretion or mutant p53 interfere with p53 function to promote chemoresistance. We highlight resistance mechanisms promoted by mutant p53 and enforced by the microenvironment, such as extracellular matrix remodeling and adaptation to hypoxia. Alterations of wild-type p53 extracellular function may create a cascade of spatial amplification loops in the tumor tissue that can influence cellular behavior far from the initial oncogenic mutation. We discuss the concept of chemoresistance as a multicellular/tissue-level process rather than intrinsically cellular. Targeting p53-dependent crosstalk mechanisms between cancer cells and components of the tumor environment might disrupt the waves of chemoresistance that spread across the tumor tissue, increasing the efficacy of chemotherapeutic agents.

## 1. Introduction

Drug resistance or chemoresistance is a hallmark of many advanced tumors and one of the main reasons for therapy failure despite developing novel compounds. Tumors may resist antineoplastic treatment either from the onset of therapy or acquire resistance later after an initial response. Subpopulations of cancer cells coexist within tumors exhibiting variability in protein expression, differentiation, proliferation, and metabolic states. This concept is known as intratumoral heterogeneity, which contributes to differential drug responses and the selection of resistant subclones during treatment [1,2]. Similarly, non-malignant cells (e.g., fibroblasts, recruited immune cells, and tissue-specific cells) and other structures (e.g., blood vessels, extracellular vesicles (EVs) and the extracellular matrix (ECM)) are variable in profile and distribution across the tumor mass, thus adding another layer of intratumoral heterogeneity. These elements compose the tumor microenvironment (TME), increasingly recognized as an essential modulator of cancer cell response to drug therapy [3,4,5]. 

Chemotherapy impacts the dynamics of cancer-associated fibroblasts (CAFs) and immune cells. Their respective subpopulations change throughout therapy both in profile and quantity, thus shaping the inflammatory microenvironment that will promote chemoresistance across the tumor [6,7]. Notably, structural components of the TME are also dynamically modulated by tumor progression and therapy. Tumor-associated ECM remodeling creates progressively stiffer ECM clusters that reduce local drug bioavailability and induce aggressive, more resistant cellular phenotypes through mechanosignaling [8,9,10,11]. In addition, the aberrant disposition of blood vessels within the tumor establishes nutrient and oxygen gradients that will directly shape metabolic heterogeneity—for instance, resistance to temozolomide in glioblastoma (GBM) cells has been shown to depend on the distance of cells from vasculature [12]. It is becoming increasingly clear that the response to chemotherapy within the same tumor is not homogenous and is in part dictated by the complex interplay between cancer cells and the TME in which they reside. 

Cell-autonomous (or intrinsic) mechanisms of chemoresistance are well understood. These comprise transmembrane transporter upregulation, hyperactivity of detoxifying enzymes, aberrant signaling of compensatory pathways, among others [13]. However, in vivo, cancer cells establish intimate communication with their TME, promoting intercellular communication and exosomal release into the microenvironment. Each layer of interaction modulates the response of cancer cells to chemotherapy. Thus, the microenvironmental changes modulated by cancer cells such as hypoxia, aberrant vascularization, and ECM abnormalities may affect drug bioavailability and cellular sensitivity to therapy within the tumor, being collectively termed “TME-driven cell-nonautonomous” (or extrinsic) mechanisms of resistance [14]. Indeed, at the most basic level of in vitro modeling, co-culturing of tumor cells with stromal cells alone is sufficient to reduce the efficacy of chemotherapeutic agents [15,16,17]. Such mechanisms have been described in multiple cancer types [5,14,17,18]. Therefore, the response of tumor tissues to chemotherapy is the result of several layers of complex and dynamic crosstalk among normal and cancer cells and their respective niches.

Several studies have recently elucidated the extrinsic mechanisms of chemoresistance and their clinical relevance. For example, a hypoxic microenvironment promotes exosomal efflux of cisplatin in ovarian cancer cells, with cisplatin-resistant patients presenting increased serum exosome concentration compared to sensitive and treatment-naïve patients [19]. Beyond drug efflux, exosome secretion increments the transit of pro-oncogenic factors between cancer cells, with benefits for their survival [20]. Thus, local TME-driven feedback loops (in this case, initiated by hypoxia) can promote waves of chemoresistance across the tumor. Along with hypoxia, ECM remodeling is another critical feature of the TME during tumor progression. Glioblastoma cells in 3D scaffolds with elevated hyaluronic acid content—an ECM alteration associated with glioblastoma progression—promoted temozolomide resistance through mechanosensitive inhibition of pro-apoptotic protein expression [21]. 

Importantly, as the TME dynamically shapes tumor response to therapy, so do tumor cells actively recruit microenvironmental components to support their features, chemoresistance included [17,18,19,20,21,22,23,24]. Such studies demand a shift in scales of our understanding of the acquisition of drug resistance, from cellular to a multicellular/tissue-level process in which the tumor-stroma crosstalk comes into play. Identifying the regulators of cell-environment interactions might unveil targetable TME-specific vulnerabilities that sensitize the whole tumor tissue to therapy, thus preventing the selection of resistant subpopulations that can reinitiate tumor growth. In this context, common mutations in tumor suppressors resulting in aberrant extracellular signaling are candidates for investigating TME-specific drug resistance [25]. 

*TP53*, which encodes the p53 transcription factor, is the most frequently altered gene across human tumors, as identified by mutation frequency analysis of The Cancer Genome Atlas Pan-Cancer cohort [26]. Usually, wild-type p53 (wtp53) levels are kept low by murine double minute 2 (MDM2)-dependent ubiquitylation and proteasome degradation. In response to cytotoxic stresses such as DNA damage, oncogene activation and hypoxia, MDM2 activity is inhibited, enabling p53 to accumulate and exert a broad transcriptional program that balances cell fate between damage repair and survival or apoptosis and death [27]. The stress response coordinated by wtp53 involves three main mechanisms that protect against tumor formation: cell cycle arrest, DNA damage repair and apoptosis induction [28]. The collection of wtp53 targets is highly context-dependent, modulating by cell type, stressor type, and stressor intensity [29]. These activities are severely dysregulated in mutant p53 (mutp53) copies, which aberrantly accumulate, lose tumor suppression function, and frequently display oncogenic gain-of-function (GOF) properties that enhance cancer cell survival [30]. 

Genome-wide analyses have firmly established the variety and functional importance of *TP53* mutations in human malignancies. They occur mainly as missense mutations, which result in full-length mutant proteins with single amino acid substitutions, or less frequent deletions and insertions in the central DNA-binding domain [28]. Mutation frequency varies widely with cancer type, from less than 5% in uveal melanoma to more than 90% in ovarian cancer [31]. Following wtp53′s role in safeguarding the genome, mutp53-harboring tumors show increased chromosomal instability and increased activation of cancer cell proliferation and survival pathways compared to wtp53-harboring tumors [31]. Although *TP53* is one of the most studied genes in cancer research, new GOF of mutp53 and roles of wtp53 tumor suppressor are still being found and show potential therapeutic new targets. Recently, Escobar-Hoyos and colleagues [32] have shown that mutp53 is associated with specific RNA splicing of GTPase-activating proteins (GAPs) to promote oncogenic activation of KRAS. Altering GAP isoform ratios decreased oncogenic KRAS signaling and pancreatic cancer cell growth. 

Besides mutations in p53 and increased MDM2 activity, p53 signaling is also frequently modulated by MDM4 (MDMX). This protein shows structural similarity to MDM2; both inhibit p53 activity and are overexpressed in various human cancers. However, unlike MDM2, which degrades p53, this protein inhibits p53 by binding its transcriptional activation domain even when binding to MDM2 [33]. 

Notably, the *TP53* gene is known to produce at least 12 p53 isoforms, including the canonical full-length wild-type protein [34,35]. The isoforms differ in structure, stability, intracellular localization, and regulation by well-known p53 partners. For instance, the truncated Δ133p53α isoform lacks the N-terminal region, which mediates the interaction with MDM2, and therefore is not subjected to MDM2-dependent proteasomal but autophagic degradation [36]. Expression patterns of the p53 isoforms also vary across tissues and with aging and disease, thus implying specific roles in the regulation of p53 activity [37]. Δ133p53α inhibits the transcriptional activity of wtp53 [34] and, in contrast to full-length wtp53, has been shown to promote pro-tumoral angiogenesis and tumor growth in glioblastoma models [35]. In contrast, another isoform, Δ133p53β, promotes invasion in breast and colon cancer cells [38]. Therefore, even a wtp53-expressing cell may harbor oncogenic p53 signaling while inhibiting wtp53 transactivation and pro-apoptotic functions. 

In addition to its role in intracellular pathways, wtp53 promotes a tumor-suppressive microenvironment by regulating the cellular secretome [39,40]. This secretome modulates intercellular communication and interactions with the surrounding microenvironment, thus extending p53′s actions beyond the plasma membrane. The p53-associated secretome modulates microenvironmental parameters, such as pH, vascularization, and the ECM, and can affect the behavior of neighboring cells [41,42,43]. In the context of cancer, mutp53 and pro-tumoral p53 isoforms have been shown to promote the tumor-stroma crosstalk that will further support tumor growth [44]. In prostate cancer and glioblastoma, for example, high levels of expression of Δ133p53β are linked to hypoxia and the shaping of an immunosuppressive and chemoresistant microenvironment [45,46], thus recapitulating phenotypes associated with functional loss of wtp53 in other studies [47,48]. The TME is deeply affected by p53 status, leading to tumor immunosuppressing or immunocompetent scenarios accordingly. Reports on the cell surface and secretome changes are instances of such influence. The work by Vogiatzi and colleagues [49] shows that mutp53 induces relevant cell surface changes by inducing the expression of ectonucleoside triphosphate diphosphohydrolase 5 (ENTPD5). ENTPD5 is an enzyme involved in forwarding N-glycosylated proteins to the Golgi apparatus. Its upregulation results in a remarkable change of cell surface tyrosine-kinase receptors (RTKs), which comprise a class of molecules heavily involved in growth factor signaling. ENTPD5 is typically expressed via AKT/PI3K pathway. However, this study reveals that mutp53 promotes ENTPD5 expression in a non-canonical pathway by docking onto the Sp1 transcription factor in the *ENTPD5* promoter region. Interestingly, ENTPD5 overexpression directs tumor cells to the same outcome as mutp53 expression. Mutp53-associated ENTPD5 overexpression was also observed in patient tumor samples, indicating the relevance of this relationship in TME modulation and, ultimately, in malignancy.

In addition to cell surface changes, p53 also plays a role in the TME secretome. Blagih and colleagues [47] have shown that p53 loss in pancreatic tumors leads to suppression of antitumoral immune response. In this study, p53 deficiency in tumor cells shapes an immunosuppressive microenvironment via the secretion of cytokines into the TME. Increased secretion of factors such as the macrophage-colony stimulating factor (M-CSF) and others enhance infiltration of immune suppressor cells, such as regulatory T-cells and myeloid cells. Blockage of M-CSF impairs myeloid cells infiltration and, consequently, leads to activation of cytotoxic T-cells in p53-deficient tumors. However, regulatory T-cells remain present and exert their immune-modulatory effect on the TME. Therefore, despite the cascade effects of myeloid cells in the TME, p53 directly mediates regulatory T-cell suppressing activity by mechanisms yet to be understood [50].

Of note, failure in tumor suppression mechanisms must occur in both the malignant and stromal compartments of developing tumors. Altered p53 functionality has been found in non-small cell lung CAFs without genetic alterations in *TP53* [51]. In these cells, some p53 protein copies adopted a “mutant-like” structural conformation that switched their transcriptional program from tumor suppression to tumor promotion, leading to the secretion of pro-oncogenic factors by CAFs. Another form of structural quenching of normal p53 function is the aggregation of mutp53 with wtp53 in cytotoxic oligomers that can be released into the microenvironment upon cell death and incorporated by other cells [52,53,54,55]. Overcoming the need for mutations to dysregulate the p53 pathway could enable mutp53 GOF properties to spread from mutp53-harboring cancer cells to wtp53-harboring cancer and stromal cells in the TME. Interestingly, increased formation of mutp53 aggregates has been associated with temozolomide resistance in glioblastoma cell lines [56], suggesting that release of structurally dysfunctional p53 into the TME might spread chemoresistance across the tumor. 

The p53 protein is uniquely poised to modulate TME-tumor cell crosstalk in response to anticancer treatment. It is a prime candidate for investigating extrinsic mechanisms of acquired chemoresistance that could be targeted to improve clinical prognoses. Mechanisms of resistance against anticancer agents that are potentially related to mutp53 expression in cancer cells have been discussed elsewhere [57]. This review will focus on GOF mechanisms acquired from *TP53* mutations that promote TME-driven chemoresistance. First, we briefly discuss the extracellular role of wtp53 in tumor suppression, which contributes to shaping a tumor-suppressive microenvironment. Then, we highlight selected TME phenomena and components (namely the secretion of pro-tumoral factors, hypoxia, cancer stem cells, and the ECM) regulated by mutp53 GOF mechanisms. We speculate how mutp53 coordinates the remodeling of plasma membrane receptors and thus controls the response of cells to the changing TME. We will finally propose that, if, on the one hand, mutp53 promotes the rewiring of intracellular signaling, on the other hand, the consequences extend beyond the plasma membrane and across the tumor tissue where mutant and non-mutant cells coexist. In this way, mutp53 reverses the tide of wtp53-associated tumor suppression with a pro-tumoral, chemoresistance-promoting wave of alternated p53 activity.

## 2. Keeping the Tissue in Check: Extrinsic Function of Wild-Type p53

The best-known tumor suppression roles of p53, such as the induction of differentiation or senescence, balance the fate of the stressed cell between adaptation or cell death. The wide range of intracellular pathways driven by p53 during damage response has been thoroughly dissected over the years. Briefly, intracellular stress signals, such as DNA damage and oncogene activation, and extracellular stress signals, such as hypoxia/anoxia and acidification, converge through multiple pathways to activate p53 or inhibit MDM2 [27]. Different stress inputs activate different pathways; for example, DNA damage repair mediated by p53 is initiated by Ataxia Telangiectasia Mutant (ATM) activation following double-strand breaks [58]; meanwhile, the cellular response to hypoxia is coordinated by hypoxia-inducible factor 1 (HIF-1) signaling, which also results in p53 stabilization [29,59,60,61]. In this way, wtp53 prevents multiple stress signals from inducing malignant transformation, although preserving wtp53 expression and related stress response pathways in cancer cells seem to promote therapy resistance in specific contexts [62].

In addition to intracellular mechanisms of p53-dependent tumor suppression, extrinsic mechanisms are evident in many uprising studies in which mouse models with different *Trp53* statuses were compared in their tumorigenic capacity. For instance, increased tumorigenesis was seen when MCF7 mammary epithelial adenocarcinoma cells were injected into p53-null mice compared to wtp53 mice [63]. In similar experimental designs, B16F1 [64] and B16F0 [65] melanoma cells injected into p53-null mice also yielded accelerated tumor growth with an increased immunosuppressive profile compared to wtp53. Compared to other genetic backgrounds, p53 knockout significantly accelerated urothelial [66] and prostate [67] tumorigenesis. Importantly, TME-associated fibroblasts present lower levels or even loss of p53 expression when compared to normal stromal cells, which correlates with accelerated tumor growth and chemoresistance to different chemotherapeutic agents, such as vincristine and cisplatin [67,68,69,70]. 

p53 suppresses the tumor microenvironment in several ways, mainly through regulation of the cellular secretome. The p53-dependent secretome has been shown to kill cancer cells. For example, activation of p53 in liver cells results in elevated sex hormone-binding globulin (SHBG) secretion, which enhances apoptosis in breast cancer cells [71]. Inhibition of aberrant angiogenesis and increased immune infiltration into pre-malignant lesions have also been related to the p53-dependent secretome, resulting in restriction of tumorigenesis [72]. Moreover, oncogenic stresses are known inductors of p53-dependent senescence [73] and the senescence-associated secretory phenotype (SASP) that promotes a tumor-suppressive microenvironment: stroma-associated SASP instructs recruited immature myeloid cells and myeloid-derived suppressor cells to express a tumor-inhibiting behavior [39]. However, persistent secretion of SASP components can have pro-tumorigenic effects, especially when normal p53 function is lost (see below).

## 3. Turn of the Resistant Tide: Mutant p53’s Nonautonomous Gain-of-Function

Substantial biochemical and biomechanical alterations occur in the ECM during cancer progression, modulating TME. The secretion of soluble factors and extracellular vesicles (EVs) is the most fundamental mechanism that shapes the TME and the TME-cell crosstalk and is also able to modulate the ECM in both the local microenvironment and distant organs, thus priming pre-metastatic niches even before the arrival of circulating tumor cells [24]. Mutations in *TP53* have a well-established role in the microenvironment by altering the normal cellular secretome, which involves aberrant paracrine/autocrine signaling, ECM remodeling, stromal cell recruitment and activation, among other mechanisms (Table 1). Importantly, mutp53 can be packed within EVs and thus, mutp53 GOF activities are communicated to neighboring cells and distant tissues through EV transfer and circulating vesicles, which together promote tumor progression and metastasis [74]. This section highlights mutp53 GOF that modulate the TME and promote TME-driven chemoresistance.

### 3.1. A Darker Side to SASP

As outlined above, wtp53 drives senescence in response to multiple stress signals while restricting SASP activity, particularly its pro-tumorigenic potential [43,79]. SASP has been shown to promote chemoresistance through the downstream activation of inductors of resistant phenotypes, such as signal transducer and activator of transcription 3 (STAT3) [80]. However, what happens when senescence is induced in a wtp53-deficient background? Coppé and colleagues demonstrated that experimental inactivation of p53 followed by induction of senescence in normal fibroblasts and prostate epithelial cells leads to SASP amplification and increased secretion of pro-tumorigenic factors [43]. Senescent, p53-deficient prostate cancer cell lines expressed similarly amplified SASP profiles. Thus, alteration of normal p53 function in both stromal and transformed cells can release a pro-oncogenic SASP into the TME that, among other biological implications, can promote chemoresistance in neighboring cells. 

Interleukin-6 (IL-6) and interleukin-8 (IL-8) are among the prominent factors identified in the amplified SASP of p53-deficient cells [43]. Accordingly, activating wtp53 with MDM2 inhibitors blunts IL-6 secretion by normal senescent fibroblasts, which reduces their ability to promote in vitro breast cancer cell invasion via SASP [81]. Significantly, stroma-derived IL-6 has been associated with resistance against cytotoxic agents in many cancer types [82,83,84]. Thus, senescence-inducing genotoxic therapy might paradoxically promote chemoresistance in a p53-deficient background by stimulating the secretion of pro-oncogenic cytokines. In addition, TME-derived IL-6 can induce epithelial-to-mesenchymal transition (EMT), a phenotypic switch that promotes chemoresistance (see below). Indeed, p53 activation suppressed the induction of the EMT marker vimentin mediated by fibroblast SASP in breast cancer cells [81]. SASP can also induce stem-like states associated with stemness markers in several contexts [85]. Therefore, SASP induction may impact patient outcomes, as stem-like cancer cells are highly chemoresistant (see below).

The genotoxic stimuli activate p53 and consequently SASP, which is restricted while wtp53 expression is retained. However, functional alteration of wtp53 arising in either the transformed or stromal compartments of the TME might exacerbate SASP. It could lead to the acquisition of chemoresistance along the course of cytotoxic therapy through SASP amplification (Figure 1). Whereas the loss of p53 in the TME unleashes the pro-tumoral role of SASP—chemoresistance included—wtp53 signaling restoration might function as a potential therapeutic strategy to attenuate such effects. However, SASP effects on chemoresistance are highly context-dependent, and wtp53 restoration should be carefully considered to prevent detrimental senescence activation [44].

Moreover, additional care is required when designing the restoration of the p53 function since it is affected by several oncogenic pathways, such as Ras mutation. In Ras-driven epithelial tumors, DNA damage induces p53 in a heterogeneously way [86]. High p53-expressing clones stimulate intercellular signaling by JAK/STAT cytokines. These cytokines supported nearby low p53-expressing surviving clones, leading to faster tumor re-establishment after therapy. These results suggested that wtp53 may contribute to tumor recovery after therapy in combination with oncogenic pathways. By interfering with this cell-cell communication loop, re-sensitization of mutant Ras tumor cells to irradiation may happen [86]. 

### 3.2. Hypoxia: p53 as Friend or Foe

Hypoxia is one of the critical microenvironmental signals that promote chemoresistance [6,14]. As the tumor grows, microenvironmental oxygen becomes limiting and hypoxic niches appear, often harboring aggressive and resistant cancer cell subpopulations that can support disease relapse. The main intracellular effector of hypoxia is the hypoxia-inducible factor 1 (HIF1) transcription factor, comprised of HIF1α and HIF1β subunits. HIF1 orchestrates intrinsic and extrinsic mechanisms of adaptation that are also known to promote chemoresistance, including metabolic reprogramming, autophagy, angiogenesis, and ECM remodeling [83,87,88,89,90]. The role of p53 signaling in hypoxia has conflicting evidence, but as in senescence, wtp53 expression is thought to dampen the most detrimental effects of HIF1 activation.

Normally, HIF1 and wtp53 antagonize each other in cancer cells [61,91,92]. For instance, HIF1-dependent inhibition of wtp53 was seen in ovarian cancer cell lines grown under hypoxia, which displayed resistance against cisplatin and paclitaxel [87]. In this context, p53 accumulated, but did not exert its pro-apoptotic program. This functional uncoupling was reversed upon HIF1 knockdown, which sensitized cells to the chemotherapeutic agents. As hypoxia develops in the extracellular space, p53 repression mediated by HIF1 can eventually establish TME-driven chemoresistance in the growing tumor. Conversely, wtp53 is able to block downstream HIF1 signaling, which seems to be partly dependent on p53-regulated microRNAs (miRNAs). In colon cancer (CRC) cells, miR-107 was identified as a direct wtp53 target that suppresses HIF1β expression and inhibits hypoxia-induced angiogenesis through blocking of vascular endothelial growth factor (VEGF) secretion [91].

Interestingly, miR-34a, another p53 target, is directly repressed by HIF1α in p53-deficient CRC cells under hypoxia [61]. These cells then undergo HIF1-induced EMT and resistance against 5-fluorouracil (5-FU); however, p53-proficient cells under the same conditions effectively upregulated miR-34a expression, which was sufficient to inhibit EMT and allowed an appropriate response to 5-FU [93]. Collectively, these studies showcase the role of wtp53 in providing a brake against mechanisms of chemoresistance that are promoted by the hypoxic TME, such as apoptosis resistance and tumor neovascularization. 

Remarkably, studies tackling hypoxia-driven chemoresistance through targeting of HIF1α were successful in sensitizing hypoxic cancer cell lines to cisplatin and 5-FU treatments only in wtp53-, but not mutp53-expressing fibrosarcoma and gastric cancer cell lines, respectively [59,93]. As outlined above, some studies show that mutp53 has the exact opposite role to wtp53 in hypoxia, actively cooperating with HIF1 to maintain the TME. In this way, mutp53 participates in a feedback loop that promotes hypoxia-driven chemoresistance in the TME in response to the hypoxic TME itself. Targeting of HIF1-mutp53 cooperation concomitant to chemotherapy has the potential of disrupting chemoresistance in hypoxic niches, increasing the efficacy of the treatment in mutp53-harboring tumors.

### 3.3. CSCs and EMT: Gearing Up for TME-Driven Chemoresistance

The emergence of intrinsically chemoresistant phenotypes within the tumor is highly dependent on TME signals. The chemoresistant phenotype regulated by the TME is perhaps best exemplified in cancer stem cells (CSCs), a subpopulation of cancer cells that reside in dedicated niches and express stem-like properties such as self-renewal and differentiation potential [94]. CSCs display low levels of proliferation, increased drug efflux, and highly competent DNA damage repair activity, resulting in intrinsic chemoresistance [95]. CSCs have been implicated in therapy failure and worse patient outcomes for many cancer types [96,97,98]. CSCs exploit multiple TME signals to promote their phenotype: stroma-derived factors, including IL-6 and transforming growth factor-β (TGF-β), were shown to support CSC niches and CSC features [99,100,101]. Importantly, tumor-associated EMT induction, which is prominently dependent on microenvironmental signals, regulates CSC formation from more differentiated cancer cells and promotes chemoresistance [100,101,102,103,104,105]. Thus, CSC and EMT phenotypes are at the heart of TME-driven chemoresistance.

Wtp53 acts as a negative regulator of both CSCs and EMT to prevent malignant transformation, mainly through downstream effector miRNAs [106,107,108]. This repressive role is reversed in mutp53, which has been shown to trigger EMT and CSC formation, thus conferring enhanced cellular plasticity [75,109]. An increase in the CSC pool promoted by mutp53 also means increased intratumoral heterogeneity and consequently chemoresistance. Heterogeneity is essential for tumors to withstand therapeutic challenges and preserve a residual number of CSCs that seed disease relapse. It is clear that once the brakes imposed by wtp53 function are removed, CSC/EMT-associated states will promote each other, a phenomenon that can be accelerated by mutp53 [107]. How mutp53 cooperates with TME signals to sustain chemoresistance in CSCs is still poorly understood. However, a recent report on therapy-induced senescence implicated this p53-dependent process in yet another mechanism through which senescence might promote TME-driven chemoresistance [110]. Doxorubicin-induced senescence in B-cell lymphoma cells increased the expression of stemness markers, which persisted high even after senescence exit was promoted by switching off p53 expression and removingdoxorubicin. Stem-like features, including increased tumorigenicity and plasticity between CSC and non-CSC states, were also observed. Thus, it is plausible that an escape from senescence driven by functional inactivation of p53, combined with mutp53-dependent SASP, could promote chemoresistance and disease relapse by leading to the emergence of CSCs within the tumor [110].

### 3.4. ECM Remodeling and Integrin Expression: Survival of the Stiffest

The ECM is a critical biophysical component of the TME and participates in most of the aforementioned processes [6]. As tumors grow and become more aggressive, the TME-associated ECM becomes stiffer, denser, and more fibrotic (or desmoplastic), reducing perfusion and impairing drug distribution within the TME, thus promoting chemoresistance. This increased matrix rigidity is associated with increased epithelial-mesenchymal transition (EMT) and resistance to chemotherapeutics in several types of cancer [10,111,112,113]. For instance, increased deposition of the ECM glycoprotein fibronectin stiffens the local ECM and is an important promoter of brain tumor growth [112]. A stiffer ECM activates oncogenic mechanosensitive signaling that drives responsiveness to therapy, as shown in the stiffness-dependent chemoresistance to doxorubicin in breast cancer cells, which could be partly explained by stiffness-induced EMT [113]. Interestingly, fibronectin expression increased 12 times when p53 expression is reduced and only tenfold when completely abolished in astrocytes from heterozygous and knockout mice for p53 (*Trp53*), respectively (Souza and Borges, unpublished results). 

The ECM interacts with the cell by integrin receptors involved in cell adhesion and transducing ECM-derived information into intracellular signaling. Several integrin receptors have been implicated in malignant phenotypes. Integrins represent a direct link between ECM composition and cancer cell response to changes in the TME [9,114]. Integrin receptors are heterodimers composed of α and β subunits, which can be paired into at least 24 known combinations that compose the integrin receptor family [115]. The combination determines the ECM ligand specificity of the receptors.

Upregulation of integrin receptor or subunit expression at the plasma membrane is a mutp53 GOF mechanism present in multiple cancer cell types (Table 2). Mutp53 proteins increased endocytic recycling of fibronectin-binding integrin α5β1 to the plasma membrane, which enhanced in vitro random migration and invasion through fibronectin-containing Matrigel [116]. Mutp53 affected integrin α5β1 by sequestering p63 protein, a transcription factor member of the p53 family [116]. This study highlights a signaling axis promoted by some p53 mutations that hijack the endocytic trafficking machinery and directly remodel the plasma membrane to drive malignant phenotypes. ECM ligands can undergo internalization mediated by endocytic recycling of their receptors, resulting in matrix turnover. Integrin α5β1 recycling has been shown to promote fibronectin internalization and turnover on the surrounding matrix [117]. Thus, under the control of p53 mutations, remodeling of ECM receptors at the plasma membrane can also directly promote remodeling of the surrounding matrix and shape a pro-invasive microenvironment.

Mutp53 can be released within tumor-derived extracellular vesicles capable of influencing the formation of pre-metastatic niches in distant organs. Such influence may be exerted by, for instance, targeting the regulation of ECM deposition by normal fibroblasts [76,126]. More recently, this mechanism for mutp53-driven remodeling of the microenvironment has been elucidated [76]. In this study, in concordance to the ones mentioned above, expression of the p53 R273H mutation led to increased recycling of integrin α5β1 at the plasma membrane of non-small cell lung carcinoma H1229 cells. Moreover, this mutp53 GOF behavior was communicated to other tumor cells and fibroblasts via the release of mutp53-containing exosomes by the R273H-expressing cells, demonstrating that such mechanism can spread locally in the tumor microenvironment and promote widespread ECM remodeling. Indeed, treatment with exosomes derived from R273H-expressing tumor cells increased integrin recycling and altered the 3D organization of the ECM deposited by fibroblasts. Finally, in pancreatic adenocarcinoma in vivo model, expression of the same p53 mutation in the primary tumor was associated with similar ECM alterations in the lung in the absence of any established metastases. Thus, aberrant integrin trafficking modulated by mutp53 drives both local and distant pro-invasive ECM remodeling, priming metastatic niches for the establishment of new tumors.

In another context, the same integrin α5β1 has been directly linked to chemoresistance. Overexpression of the α5 subunit in glioblastoma cells prevented wtp53 activation mediated by temozolomide, leading to chemoresistance even against high concentrations of this agent [127]. Interestingly, persistent wtp53 activation achieved by the MDM2 inhibitor nutlin-3a greatly inhibited α5 expression in both glioblastoma [127] and colorectal cancer [128] cells. In glioblastoma cells, nutlin-3a treatment was shown to abolish integrin α5β1-mediated chemoresistance against temozolomide. This reveals a negative feedback loop between wtp53 and integrin α5β1 that might be lost in mutp53-expressing cells, thus allowing integrins at the plasma membrane to engage to a TME-specific ECM and trigger cancer cell survival. Indeed, modulating p53 expression in mutp53-expressing glioblastoma cells had no effect on α5 subunit expression [127]. More evidence is needed to define the relevance of integrins as therapeutic targets to increase drug sensitivity in cancer cells, and the dependence of integrin signaling on context needs to be taken into account when comparing different in vitro studies. However, increases in drug sensitivity have been seen following pharmacological inhibition of specific integrin subunits: for instance, β1 inhibition increased sensitivity to gefitinib in wtp53-harboring lung cancer A549 cells, whereas it led to only a small increase in sensitivity to cisplatin in A431 cells, which carry the R273H p53 mutation (Table 1, refs. [84,85]).

In addition to integrin expression, p53 regulates ECM deposition through the cellular secretome, which is hijacked by mutp53, resulting in tumorigenic ECM remodeling and TME-driven chemoresistance [77,129]. One mechanism for such remodeling is the enhanced secretion of matrix metalloproteinases (MMPs). In prostate cancer cell lines, Zhu and colleagues [130] have shown that p53 regulates EMMPRIN (extracellular matrix metalloproteinase inducer), an N-glycosylated plasma membrane protein upregulated in different tumor types. EMMPRIN promotes invasion and metastasis via the activation of MMPs. They observed that p53-null and mutp53 cells lines expressed higher levels of EMMPRIN compared to wtp53 tumor cells. Additionally, p53-null and mutp53 cells were more invasive, and EMMPRIN silencing hampered the invasion of p53-null cells. EMMPRIN transcriptional levels were not altered by wtp53 to mutp53 expression. However, when cells were treated with chloroquine, a lysosomal degradation pathway inhibitor, EMMPRINN protein levels in wtp53 cells reached mutp53 levels, indicating that p53 modulates EMMPRIN cell surface expression by directing the protein to lysosomal degradation. This regulation is relevant in the modulation of the TME, as EMMPRIN activates MMP9, a central metalloproteinase in many tumor types [130]. 

ECM remodeling might also lead to chemoresistance. In a mouse model of pancreatic cancer, primary CAFs expressing mutp53 deposited a stiffer ECM compared to their p53-null counterparts, evidencing a mutp53 GOF mechanism [131]. Moreover, p53-null CAFs could be instructed to deposit mut-p53-like ECM when treated with conditioned medium (CM) derived from either mutp53 pancreatic cancer cells (PCCs) or CAFs. This model highlights the importance of *TP53* status in both the transformed and stromal compartments of the TME. Remarkably, PCCs kept in contact with the ECM deposited by mutp53-expressing CAFs resisted gemcitabine-induced apoptosis, irrespective of PCC *TP53* status. On the other hand, PCCs in contact with the ECM deposited by p53-null CAFs showed cytotoxic responses to the same agent. In vivo, mutp53 CAFs were shown to significantly delay tumor response to chemotherapy, even though they coexisted with p53-null CAFs. 

Our group has previously demonstrated another role for p53 function on stromal ECM and its impact on cancer cell behavior [132]. This study analyzed the role of p53 in the crosstalk between GBM and astrocytes, a prominent cellular component of the GBM microenvironment. Primary p53-heterozygous astrocytes isolated from mice deposited an ECM enriched for fibronectin and laminin compared to their p53-proficient counterparts. Interestingly, increased laminin deposition was shown to be mutp53/HIF1α-dependent under hypoxia in non-small cell lung cancer cells [77], whereas the laminin-γ2 subunit has been implicated in resistance against docetaxel and taxane in ovarian cancer cell lines [133]. In our study, contact with the p53-heterozygous astrocytic ECM was sufficient to promote the expression of some EMT markers and resistance to spontaneous cell death in GBM cell lines, irrespective of GBM *TP53* status. Moreover, treatment with GBM-conditioned medium prevented p53 accumulation in p53-proficient astrocytes even under genotoxic stress. Conditioned medium-treated normal astrocytes deposited an ECM that promoted GBM cell survival, akin to the p53-heterozygous astrocytes. 

The observations above are consistent with the study by Vennin and colleagues and another study demonstrating a reduction of astrocytic p53 expression mediated by GBM cell-derived vesicles [134]. Altogether, these studies demonstrate the vital role that p53 extrinsic mechanisms have in regulating the tumor-stroma crosstalk and biophysical properties of the TME, which mutp53 exploits to promote TME-driven chemoresistance and cancer cell survival (Figure 1). 

## 4. Conclusions and Perspectives

Solid tumors can be considered unstructured tissues composed and regulated by cellular and non-cellular components. The concept of chemoresistance is expanding beyond the cancer cell. It emerges as a direct consequence of heterogeneity, collaboration among clusters, and microenvironment modification. It is promoted by properties that include the physical-chemical conditions of the microenvironment, secretion of extracellular factors and vesicles, and ECM deposition. p53 plays a role in many of these processes. GOF activities of mutp53 turn tumor suppression of wtp53 into tumor promotion. Consequently, the mutp53-harboring cell holds a chemoresistant microenvironment (Table 3).

Mutational events can cause a change in the structure of the p53 protein, which includes the formation and accumulation of intracellular oligomeric aggregates and fibrils that convert wtp53 into a deformed conformation [53,55]. Amyloid aggregates of mutant p53 have been detected in melanoma and breast cancer, and high levels of these aggregates seem to be related to more invasive tumors. Amyloid oligomers with high levels of wtp53/mutp53 co-aggregates have been detected in prostate cancer, leading to a dominant-negative effect on wtp53 function that can be transmitted to neighboring cells in a prion-like behavior [55]. It is crucial to investigate different strategies tackling the TME to enable a chemotherapy-sensitive state in tumor tissues. For instance, antibodies and other pharmacological “reactivators” of p53 that alter protein conformation can change the structure of mutp53 to recover the original tumor-suppressive function of wtp53 [55,135]. 

Furthermore, wtp53 in stromal cells associated with the TME may present a mutant-like conformation or altered function. Hence, treatment with agents that modulate the conformation of mutp53 to wild-type may impact both the tumor and stromal components of the TME while also boosting tumor suppression mechanisms related to wtp53. Several strategies are already trying to inhibit mutp53-related dominant-negative effects and GOF through induction of conformational change [136]. The primary idea stands to sensitize cancer cells to apoptosis-induced therapy. However, the combined intra- and extracellular effects of p53 signaling may be worth exploring to turn the TME into an environment of tumor suppression. 

In one such approach, Guo and colleagues [137] used nutlin-3a, an MDM2 inhibitor, to reactivate p53 in the TME and boost the antitumoral immune response. Nutlin-3a has been tested as a therapeutic agent via systemic injection; however, it is highly toxic. In this work, intratumoral nutlin-3a injection is used in a non-toxic concentration to induce p53 expression locally and, consequently, promote tumor immunogenic cell death, which comprises calreticulin cell surface exposure, as well as HMGB1 and ATP secretion. These factors hinder the influence of the myeloid-derived suppressor cells on the TME and allow antitumoral T-cell activation. The Nutlin-3a approach poorly affects tumors that are not typically infiltrated by T-cells, such as B16-derived melanomas. For such cases, authors describe pro-inflammatory treatments as partially effective in attracting immune cells into the TME as a combined strategy for nutlin-3a injection. 

Besides p53 signaling, an anti-cancer treatment that simultaneously targets intra- and extracellular key molecules that support tumor growth will be continually explored. Targeting the tumor microenvironment by drugs and antibodies aiming at angiogenic pathways and immune cells within the microenvironment are two strategies in progress. However, identifying relevant biomarkers is still under development to help recognize patients who could benefit from immune checkpoint blockade therapy. Most patients do not respond to antiangiogenic treatment or develop resistance when used as a single agent. Therefore, understanding tumor biology as dynamic tissues and identifying intrinsic and extrinsic target pathways should render effective therapies.

## Figures and Tables

**Figure 1 membranes-12-00202-f001:**
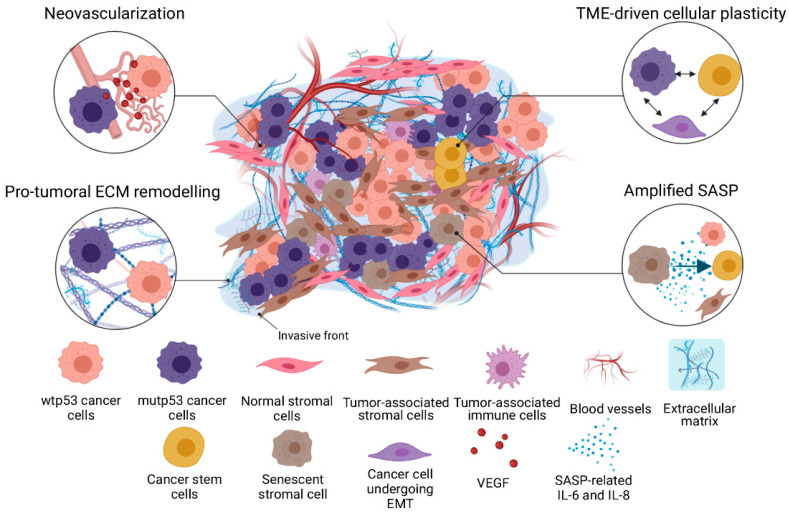
Mechanisms of TME-driven chemoresistance promoted by p53 loss. Tumor tissue-level chemoresistance is the result of complex interactions between cancer cells and their TME, with p53 acting as a key regulator. Functional loss of normal p53 can happen in tumor-associated stromal cells, such as fibroblasts, epithelial cells and tissue-resident cells, which cooperates with GOF p53 mutations in cancer cells to enforce a chemoresistant microenvironment, while also augmenting resistant and invasive phenotypes. CAF, cancer-associated fibroblast; ECM, extracellular matrix; EMT, epithelial-to-mesenchymal transition; IL-6, interleukin-6; IL-8, interleukin-8; mutp53, mutant p53; SASP, senescence-associated secretory phenotype; TME, tumor microenvironment; VEGF, vascular endothelial growth factor; wtp53, wild-type p53.

**Table 1 membranes-12-00202-t001:** Relationship between p53 mutations and tumor microenvironment.

p53 Mutation	Cancer Type	Change in Microenvironment	Reference
R273H	Pancreatic ductal carcinoma	Release of mutp53-containing	[74]
		EVs.	
	Colon carcinoma	Enhancement of CSC expansion.	[75]
	Non-small cell lung carcinoma	Pro-invasive microenvironment and	[76,77]
		ECM regulation.	
R175H	Non-small cell lung carcinoma	Pro-invasive microenvironment.	[76]
V157F	Pancreatic ductal carcinoma	Release of mutp53-containing	[74]
		EVs.	
R249S	Pancreatic ductal carcinoma	Release of mutp53-containing	[74]
		EVs.	
P309S	Colon carcinoma	Enhancement of CSC expansion.	[75]
R248W	Colon carcinoma	Enhancement of CSC expansion.	[75]
R246I	Non-small cell lung carcinoma	ECM regulation.	[77]
R248	Ovarian cancer	Increased adhesion to mesothelial	[78]
		cells.	

**Table 2 membranes-12-00202-t002:** Relationship between p53 mutations and integrin signaling in cancer cells.

p53 Mutation	Integrin Receptor or Subunit	Cancer Cell Line/Type	Resultant Phenotype	Reference
R248Q	α_V_β_3_	KYSE150 (esophageal squamous cell carcinoma)	Upregulation of integrins and downstream activation of ERK signaling.	[118]
	β_4_	OVCAR-3 (high-grade serous ovarian adenocarcinoma)	Upregulation of integrins and downstream activation of PI3K/Akt signaling.	[78]
R273H	α_5_β_1_	H1299 (non-small cell lung carcinoma)	Enhanced integrin and EGFR recycling to the plasma membrane and concomitant activation of MET signaling.	[116,119]
		H1975 (non-small cell lung carcinoma)	Enhanced integrin and EGFR recycling to the plasma membrane.	[120]
	β_1_	A431 (lung squamous cell carcinoma)	Modest cisplatin resistance related to integrin expression.	[121]
	α_V_	GBM6 (primary glioblastoma)	Upregulation of integrin expression resulting in ECM-mediated carmustin resistance.	[122]
	β_4_	HT29 (colorectal adenocarcinoma)	Loss of wtp53-dependent integrin repression.	[123]
R172H	β_1_	Pancreatic ductal adenocarcinoma cells derived from an oncogenic KRAS/mutp53 mouse model	Upregulation of integrin expression resulting in basement membrane-mediated trametinib resistance.	[124]
R175H	β_1_	SNO (human oesophageal squamous carcinoma)	Upregulation of integrin signaling resulting in sustained FAK activation and resistance to caspase-8 activation.	[125]

**Table 3 membranes-12-00202-t003:** Relationship between p53 mutations and chemoresistance.

p53 Mutation	Cancer Type	Drug Chemoresistance	Reference
R273H	Colon carcinoma	5FU, Cisplatin	[75]
	Epidermoid carcinoma	Cisplatin	[121]
P309S	Colon carcinoma	5FU, Cisplatin	[75]
R248W	Colon carcinoma	5FU, Cisplatin	[75]
Q136X	Ovarian cancer	Cisplatin, Paclitaxel	[87]
G245R	Fibrosarcoma	5FU, Cisplatin	[93]

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
