# Peer review of "p53 Signaling on Microenvironment and Its Contribution to Tissue Chemoresistance"

_membranes, 2022, doi:10.3390/membranes12020202_

Round 1
Reviewer 1 Report
The review “p53 Signaling on Microenvironment Contribution to Tissue Chemoresistance “ provides a broad look at the role of p53 tumor suppressor in shaping tumor microenvironment (TME). It is focused on non-cell-autonomous mechanisms triggered by wild-type and mutant p53 which can lead to development of tumor-supportive microenvironment and promote chemoresistance. Understanding the interactions between cancer cells and non-malignant cells composing the tumor is very important for deciphering the mechanisms of therapy resistance. Despite enormous number of publications dedicated to p53 studies, its role in regulation of TME is poorly understood. Therefore, this review which encompasses numerous recent publications in this field is timely and of great interest. The manuscript is well structured, clearly and well written. There are, however, a few issues that need to be improved.
The authors tend to cite preferably review papers. It is recommended to include more references to the original papers, this would improve the impact of this review.
Some recent publications, for example, describing the mechanisms mediating functions of mutant p53 in processing of membrane proteins or suppression of anti-tumor immunity are missing (PNAS, Cancer Cell).
Author Response
Rio de Janeiro, 31st January 2022
Special Issue, Biomembranes and Cell Signaling in Health and Disease
Manuscript ID: membranes-1561447
Dear Reviewer No 1
We have revised the manuscript entitled "p53 Signaling on Microenvironment Contribution to Tissue Chemoresistance" to fully address the Reviewers’ comments.
We appreciate the Referees’ critical and constructive comments and believe that they strengthened the manuscript.
Furthermore, we have incorporated all suggestions made by the reviewers. Those changes are highlighted within the manuscript. Please see below, in blue, for a point-by-point response to the reviewer’ comments.
- The authors tend to cite preferably review papers. It is recommended to include more references to the original papers, this would improve the impact of this review.
Reply: Original papers were cited, replacing most of the review papers.
2) Some recent publications, for example, describing the mechanisms mediating functions of mutant p53 in processing of membrane proteins or suppression of anti-tumor immunity are missing (PNAS, Cancer Cell).
Reply: Thank you for point the missing publications. We have extended or created new paragraphs to address mutant p53 mechanisms processing membrane proteins or suppressing antitumor immunity.
The new paragraphs are highlighted in the text and are shown below:
Lines 114-120: Although TP53 is one of the most studied genes in cancer research, new GOF of mutp53 and roles of wtp53 tumor suppressor are still being found and show potential therapeutic new targets. Recently, Escobar-Hoyos and colleagues (32) have shown that mutp53 is associated with specific RNA splicing of GTPase-activating proteins (GAPs) to promote oncogenic activation of KRAS. Altering GAP isoform ratios decreased oncogenic KRAS signaling and pancreatic cancer cell growth.
Lines 166-176: In addition to cell surface changes, p53 also plays a role in the TME secretome. Blagih and colleagues (47) have shown that p53 loss in pancreatic tumors leads to suppression of antitumoral immune response. In this study, p53 deficiency in tumor cells shapes an immunosuppressive microenvironment via the secretion of cytokines into the TME. Increased secretion of factors such as the macrophage-colony stimulating factor (M-CSF) and others enhance infiltration of immune suppressor cells, such as regulatory T-cells and myeloid cells. Blockage of M-CSF impairs myeloid cells infiltration and, consequently, leads to activation of cytotoxic T cells in p53-deficient tumors. However, regulatory T-cells remain present and exert their immune-modulatory effect on the TME. Therefore, despite the cascade effects of myeloid cells in the TME, p53 directly mediates regulatory T-cell suppressing activity by mechanisms yet to be understood (50).
Lines 469-484: In addition to integrin expression, p53 regulates ECM deposition through the cellular secretome, which is hijacked by mutp53, resulting in tumorigenic ECM remodeling and TME-driven chemoresistance (77,130). One mechanism for such remodeling is the enhanced secretion of matrix metalloproteinases (MMPs). In prostate cancer cell lines, Zhu and colleagues (131) have shown that p53 regulates EMMPRIN (extracellular matrix metalloproteinase inducer), an N-glycosylated plasma membrane protein upregulated in different tumor types. EMMPRIN promotes invasion and metastasis via the activation of MMPs. They observed that p53-null and mutp53 cells lines expressed higher levels of EMMPRIN compared to wtp53 tumor cells. Additionally, p53-null and mutp53 cells were more invasive, and EMMPRIN silencing hampered the invasion of p53-null cells. EMMPRIN transcriptional levels were not altered by wtp53 to mutp53 expression. However, when cells were treated with chloroquine, a lysosomal degradation pathway inhibitor, EMMPRINN protein levels in wtp53 cells reached mutp53 levels, indicating that p53 modulates EMMPRIN cell surface expression by directing the protein to lysosomal degradation. This regulation is relevant in the modulation of the TME, as EMMPRIN activates MMP9, a central metalloproteinase in many tumor types (131).
Lines 554-564: In one such approach, Guo and colleagues (138) used nutlin-3a, an MDM2 inhibitor, to reactivate p53 in the TME and boost the antitumoral immune response. Nutlin-3a has been tested as a therapeutic agent via systemic injection; however, it is highly toxic. In this work, intratumoral nutlin-3a injection is used in a non-toxic concentration to induce p53 expression locally and, consequently, promote tumor immunogenic cell death, which comprises calreticulin cell surface exposure, as well as HMGB1 and ATP secretion. These factors hinder the influence of the myeloid-derived suppressor cells on the TME and allow antitumoral T-cell activation. The Nutlin-3a approach poorly affects tumors that are not typically infiltrated by T-cells, such as B16-derived melanomas. For such cases, authors describe pro-inflammatory treatments as partially effective in attracting immune cells into the TME as a combined strategy for nutlin-3a injection.
Sincerely,
Helena L. Borges
Associate Professor
Universidade Federal do Rio de Janeiro

Reviewer 2 Report
I thank the authors for submitting the review manuscript entitled «p53 Signaling on Microenvironment Contribution to Tissue 2 Chemoresistance». The authors comprehensively describe the role of wt and particularly mut p53 on tumor microenvironment and microenvironment-related mechanisms of chemoresistance.
The review is generally very interesting and informative, well written and the main literature data are reported.
I have few remarks:
I would suggest use of more tables to summarize the events related to the impact p53 (both wt and mut) on tumor microenvironment and chemoresistance in several tumor types. That would help in coping with current knowledge.
It should be described that p53 have several isoforms, some of them with the potential to alter the tissue microenvironment upon e.g. hypoxia.
Line 255 – the sentence is covered by the picture
Line 301-302 - This sentence seems not to be related to Figure 1.
Author Response
Rio de Janeiro, 31 January 2022
Special Issue, Biomembranes and Cell Signaling in Health and Disease
Manuscript ID: membranes-1561447
Dear Reviewer No 2,
We have revised the manuscript entitled "p53 Signaling on Microenvironment Contribution to Tissue Chemoresistance" to fully address the Reviewers’ comments.
We appreciate the Referees’ critical and constructive comments and believe that they strengthened the manuscript.
Furthermore, we have incorporated all suggestions made by the reviewers. Those changes are highlighted within the manuscript. Please see below for a point-by-point response to the comments.
- I would suggest use of more tables to summarize the events related to the impact p53 (both wt and mut) on tumor microenvironment and chemoresistance in several tumor types. That would help in coping with current knowledge.
Reply: We have added two more tables to summarize p53 impact on tumor microenvironment and chemoresistance.
Old Table 1 is now Table 2. The new tables are listed below:
|
p53 mutation |
Cancer type |
Change in microenvironment |
Reference |
|
R273H |
Pancreatic ductal carcinoma |
Release of mutp53-containing EVs. |
(73) |
|
Colon carcinoma |
Enhancement of CSC expansion. |
(74) |
|
|
Non-small cell lung carcinoma |
Pro-invasive microenvironment and ECM regulation. |
(75,76) |
|
|
R175H |
Non-small cell lung carcinoma |
Pro-invasive microenvironment. |
(75) |
|
V157F |
Pancreatic ductal carcinoma |
Release of mutp53-containing EVs. |
(73) |
|
R249S |
Pancreatic ductal carcinoma |
Release of mutp53-containing EVs. |
(73) |
|
P309S |
Colon carcinoma |
Enhancement of CSC expansion. |
(74) |
|
R248W |
Colon carcinoma |
Enhancement of CSC expansion. |
(74) |
|
R246I |
Non-small cell lung carcinoma |
ECM regulation. |
(76) |
|
R248 |
Ovarian cancer |
Increased adhesion to mesothelial cells. |
(77) |
Line 264: Table 1. Relationship between p53 mutations and tumor microenvironment.
|
p53 mutation |
Cancer type |
Drug Chemoresistance |
Reference |
|
R273H |
Colon carcinoma |
5FU, Cisplatin |
(74) |
|
Epidermoid carcinoma |
Cisplatin |
(120) |
|
|
P309S |
Colon carcinoma |
5FU, Cisplatin |
(74) |
|
R248W |
Colon carcinoma |
5FU, Cisplatin |
(74) |
|
Q136X |
Ovarian cancer |
Cisplatin, Paclitaxel |
(86) |
|
G245R |
Fibrosarcoma |
5FU, Cisplatin |
(92) |
Line 529: Table 3. Relationship between p53 mutations and chemoresistance.
2) It should be described that p53 have several isoforms, some of them with the potential to alter the tissue microenvironment upon e.g. hypoxia.
Reply: We have added a paragraph for discussing the p53 isoforms.
Lines 126-138: Notably, the TP53 gene is known to produce at least 12 p53 isoforms including the canonical full-length wild-type protein (34,35). The isoforms differ in structure, stability, intracellular localization, and regulation by well-known p53 partners. For instance, the truncated Δ133p53α isoform lacks the N-terminal region, which mediates the interaction with MDM2, and therefore is not subjected to MDM2-dependent proteasomal but autophagic degradation (36). Expression patterns of the p53 isoforms also vary across tissues and with aging and disease, thus implying specific roles in the regulation of p53 activity (37). Δ133p53α inhibits the transcriptional activity of wtp53 (34) and, in contrast to full-length wtp53, has been shown to promote pro-tumoral angiogenesis and tumor growth in glioblastoma models (35). In contrast, another isoform, Δ133p53β, promotes invasion in breast and colon cancer cells (38). Therefore, even a wtp53-expressing cell may harbor oncogenic p53 signaling while inhibiting wtp53 transactivation and pro-apoptotic functions.
Lines 139-165: In addition to its role in intracellular pathways, wtp53 promotes a tumor-suppressive microenvironment by regulating the cellular secretome (39,40). This secretome modulates intercellular communication and interactions with the surrounding microenvironment, thus extending p53’s actions beyond the plasma membrane. The p53-associated secretome modulates microenvironmental parameters, such as pH, vascularization, and the ECM, and can affect the behavior of neighboring cells (41–43). In the context of cancer, mutp53 and pro-tumoral p53 isoforms have been shown to promote the tumor-stroma crosstalk that will further support tumor growth (44). In prostate cancer and glioblastoma, for example, high levels of expression of Δ133p53β is linked to hypoxia and the shaping of an immunosuppressive and chemoresistant microenvironment (45,46), thus recapitulating phenotypes associated with functional loss of wtp53 in other studies (47,48). The TME is deeply affected by p53 status, leading to tumor immunosuppressing or immunocompetent scenarios accordingly. Reports on the cell surface and secretome changes are instances of such influence. The work by Vogiatzi and colleagues (49) shows that mutp53 induces relevant cell surface changes by inducing the expression of ectonucleoside triphosphate diphosphohydrolase 5 (ENTPD5). ENTPD5 is an enzyme involved in forwarding N-glycosylated proteins to the Golgi apparatus. Its upregulation results in a remarkable change of cell surface tyrosine-kinase receptors (RTKs), which comprise a class of molecules heavily involved in growth factor signaling. ENTPD5 is typically expressed via AKT/PI3K pathway. However, this study reveals that mutp53 promotes ENTPD5 expression in a non-canonical pathway by docking onto the Sp1 transcription factor in the ENTPD5 promoter region. Interestingly, ENTPD5 overexpression directs tumor cells to the same outcome as mutp53 expression. Mutp53-associated ENTPD5 overexpression was also observed in patient tumor samples, indicating the relevance of this relationship in TME modulation and, ultimately, in malignancy.
3) Line 255 – the sentence is covered by the picture
Reply: We have corrected it.
4) Line 301-302 - This sentence seems not to be related to Figure 1.
Reply: We removed the indication of Figure 1.
Sincerely,
Helena L. Borges
Associate Professor
Federal University of Rio de Janeiro
